# Event-related theta and gamma band oscillatory dynamics during visuo-spatial sequence memory in younger and older adults

**Makenna B. McGill, Paul D. Kieffaber** *

Department of Psychological Sciences, College of William & Mary, Williamsburg, Virginia, United States of America

* pdkieffaber@wm.edu

**Data Availability Statement:** All relevant data is available from OpenNeuro at doi:10.18112/openneuro.ds004942.v1.0.0 (accession number ds004942).

## Abstract

Visuo-spatial working memory (VSWM) for sequences is thought to be crucial for daily behaviors. Decades of research indicate that oscillations in the gamma and theta bands play important functional roles in the support of visuo-spatial working memory, but the vast majority of that research emphasizes measures of neural activity during memory retention. The primary aims of the present study were (1) to determine whether oscillatory dynamics in the Theta and Gamma ranges would reflect item-level sequence encoding during a computerized spatial span task, (2) to determine whether item-level sequence recall is also related to these neural oscillations, and (3) to determine the nature of potential changes to these processes in healthy cognitive aging. Results indicate that VSWM sequence encoding is related to later ($\sim$700 ms) gamma band oscillatory dynamics and may be preserved in healthy older adults; high gamma power over midline frontal and posterior sites increased monotonically as items were added to the spatial sequence in both age groups. Item-level oscillatory dynamics during the recall of VSWM sequences were related only to theta-gamma phase amplitude coupling (PAC), which increased monotonically with serial position in both age groups. Results suggest that, despite a general decrease in frontal theta power during VSWM sequence recall in older adults, gamma band dynamics during encoding and theta-gamma PAC during retrieval play unique roles in VSWM and that the processes they reflect may be spared in healthy aging.

## 1. Introduction

Visuo-spatial working memory (VSWM) is a core function of the brain that permits the short-term maintenance and manipulation of information related to an object's spatial location and/or orientation [1, 2]. Originally conceived as a subcomponent of a domain-general working memory system that, as the name implies, temporarily stores the visuo-spatial information, as opposed to verbal information [3], VSWM has historically been further subdivided into

**Funding:** The authors received no specific funding for this work.

**Competing interests:** The authors have declared that no competing interests exist.

distinct components related to (1) the recall of shapes and texture (i.e., visual) and (2) the recall of spatial location and serial order, including spatial sequences [4]. Visual sequences tend to include novel stimuli that are presented in the same region in space, while spatial sequences usually consist of visually identical items that occupy different areas of the visual field. While VSWM encompasses both types of stimuli, these two components may place unique demands on sequence memory [5, 6].

The ability to process serial order is recognized as a fundamental component of cognition across sensory domains [7]. When measured in the visual domain, serial order processing is sometimes referred to as "visuo-spatial sequence memory" or "serial spatial recall" [8]. This kind of VSWM for sequences is thought to be crucial for daily behaviors such as planning, learned motor skills, and goal-directed action, including imitating sequences of actions observed in others [9].

Measures of VSWM for spatial sequences have long been used in clinical neuropsychological assessments and experimental research, such as the Corsi block-tapping task [10] and the Spatial Span subtest of the Wechsler Memory Scale–Third Edition (WMS-III) [11]. In the original Corsi block-tapping task, the experimenter touches nine wooden blocks in a sequence. The participant is then tasked with reproducing the sequence of taps, and the number of correct replications (correct blocks in correct sequence) is used as an index of the individual's VSWM capacity, or "span" [12]. Similarly, the WMS-III Spatial Span task uses ten blocks, with participants asked to reproduce a series of taps either in order or in reverse order.

Computerized versions of these tasks have been used in conjunction with neuroimaging methods to study the encoding, retention, and/or recall of visuo-spatial sequences in memory. However, much of the previous work used visual sequences that lacked the spatial component of VSWM [13, 14]. Other work has focused on the *retention* period [15, 16], which does not involve the presentation (i.e., *encoding*) or reproduction (i.e., *recall*) of stimuli and therefore does not lend itself to time-locked, sequence position-level analyses. Furthermore, the few studies that include encoding and recall periods of VSWM compare short and long memory sequence loads [8, 17], rather than analyze each item in the sequence. This has left a gap in the literature regarding the position-level analysis of the encoding and recall of visuo-spatial sequences in working memory.

## 1.1 Overview of electrophysiological correlates of VSWM sequences

Findings from electroencephalography (EEG) and magnetoencephalology (MEG) studies suggest that oscillatory power in theta (4–7 Hz) and gamma (25–75 Hz) bands is associated with successful episodic encoding and recall [18, 19]. There is also evidence that sequence memory may rely on interactions between the amplitude of gamma oscillations and the phase of theta oscillations [20, 21]. For example, research suggests that the learning and memory performance are related to the strength of phase-amplitude coupling (PAC) in the theta and gamma bands [22, 23]. These theta-gamma interactions during sequence memory have been associated with the hippocampus and surrounding medial temporal lobe [14, 24]. Furthermore, visuo-spatial memory has been linked to spectral power in both theta and gamma band oscillations [25, 26], and the encoding of visuospatial information has frequently been localized to the frontal and parietal cortices [27–29].

Theta power has been shown to increase with greater working memory load [30–32] and with greater task difficulty (i.e., easy vs. hard) [33]. Similarly, previous work has found increases in gamma power with greater working memory load [24, 34–36]. While the extant literature favors the link between gamma band oscillations and successful working memory encoding during visual recognition tasks [37, 38], theta band oscillations have also shown

increased activity during successful encoding of visual stimuli compared to the encoding of items that are later forgotten [37–39]. A recent time-frequency analysis identified low-frequency Event-Related Spectral Perturbation (ERSP) increases during the encoding phase of a visual working memory task [40]. Investigating neural oscillations in theta and gamma bands is therefore important to furthering our understanding of successful VSWM encoding for sequences.

## 1.2 Encoding and retention of VSWM sequences

It is noteworthy that most of the extant literature on the oscillatory neural dynamics of sequence memory has focused on the retention, or maintenance, phase of memory, rather than encoding and/or recall. Within this literature, theta oscillations have been shown to increase during the retention period of *sequence* working memory trials, whereas gamma power is increased during the retention period of *spatial* working memory trials [16, 41]. Theta-gamma PAC has also been implicated in the visuospatial representation of items in working memory; during the retention period of a VSWM task, PAC was found to increase as working memory load increased [42]. These studies are informative regarding the potential differentiation of the sequence and visuo-spatial components of memory tasks, however, questions remain regarding these oscillatory dynamics during encoding and recall.

Neuroimaging studies that measured stimulus-locked activity at the time of encoding provide some insight about the encoding of individual items presented in a visual sequence. In one study using MEG, participants were tested for the temporal order of a 6-object sequence. While gamma power did not change throughout the encoding of the sequence, theta-gamma PAC was found to decrease across items in the sequence due to a widening of the gamma power distribution over the theta phase. The authors propose that encoding of sequences is supported by a temporal shifting of gamma power with each subsequent item along the phase of theta [14]. In contrast, another study that employed MEG recordings and a spatial sequence ranging from 2 to 4 items in length found that both theta and gamma power increased with memory load during sequence retention. Notably, while the theta power increase was limited to the retention interval, the increase in gamma power was also observed during sequence encoding [17]. Despite the recent emergence of research designs that permit the measurement of oscillatory dynamics that are time-locked to stimulus encoding, further research is needed in order to determine how oscillatory dynamics are related to differences across serial positions in spatial sequences in VSWM.

## 1.3 Recall of VSWM sequences

Even less is known about the role of oscillatory dynamics in the recall processes of VSWM, particularly for sequence memory, most likely due to the inherent challenge of identifying reliable time-locking events to mark the initiation of recall. Previous research has found that hippocampal activity patterns reflect temporal position during recall of a 5-object visual sequence [43]. However, behavioral studies of the Corsi block-tapping task show that initial item response times are slower for long sequences (6 or more items) compared to short sequences (3–4 items), suggesting that serial order processing occurs before recall responses are made [44].

One study using a delayed-match-to-sample task suggests that theta amplitude decreases with increasing memory load during visual working memory recall [45], but this study did not address recall of spatial sequences. Although some studies have focused on serial position analysis and clustering effects during spatial sequence recall, their analyses are often limited to comparisons between short and long sequences [8], and/or do not include electrophysiological measures [46–48]. However, one study using MEG to investigate the encoding, retention, and

recall of a spatial sequence ranging from 2 to 4 items suggests that increases in both theta and gamma power can be observed across all phases of memory but that these increases are largest during the recall phase [17], underscoring the potential value of improving our understanding about the recall processes.

In one related study using EEG and task stimuli consisting of either fixed or random sequences of 5 objects, response-locked frontal theta power was observed to be decreased during recall of learned, fixed sequences compared with random sequences, but only for the later positions (3–4) of the sequence [13]. Frontal theta power also increased after recall of the last position (5), suggesting that theta power may be inversely related to the prediction of upcoming items in sequence. Crivelli-Decker and colleagues [13] further demonstrated that single trial theta oscillations were predictive of reaction time for future objects in fixed sequences, further supporting the role of theta oscillations in visual sequence recall. Prior work indicates that theta-gamma PAC increases during the cued recall of learned spatial representations [49] and that PAC is increased when temporal ordering is required on an N-back task [50], however, it remains unclear if changes in PAC occur across serial position during the recall of spatial sequences. While previous literature suggests the importance of theta and gamma oscillations in the accurate encoding, retention, and/or recall of visual and/or temporal information, the primary aim of the present study is to determine how oscillatory dynamics in the theta and gamma frequency bands vary as a function of sequence position for correctly recalled items in a *spatial* visual sequence.

## 1.4 VSWM and sequence memory in older adults

VSWM and sequence memory tasks like the Corsi block-tapping task and other visuo-spatial sequence tasks are commonly used as neuropsychological measures [12] of cognitive changes and show reliable sensitivity to age-related changes in memory performance [51–54]. A meta-analysis concluded that age-related performance differences are greater for context memory, particularly spatio-temporal context, than for content memory, further supporting the notion that spatial sequence memory may be particularly vulnerable to normal aging [55]. While impaired behavioral performance on VSWM and/or sequence memory tasks is one indicator of age-related changes, neuroimaging methods support the notion that these changes are rooted in altered brain activity during the encoding and recall of spatial sequences.

EEG studies have shown that, compared to younger adults, older adults demonstrate reduced theta power during spatial working memory tasks [56], particularly with increasing task difficulty [57], and show age-related changes in the activity of fronto-parietal networks [58, 59]. Research also indicates that sequence memory is susceptible to aging processes [60, 61], though it remains largely understudied. EEG findings on frontal theta power in older adults during visual n-back tasks are mixed; one study found that frontal theta power increased with task difficulty and was more pronounced in older adults compared to younger adults [62], while a another study found reduced frontal theta power in older adults [63]. Additionally, older adults were found to have decoupled theta-gamma rhythms during the maintenance period of a visual change-detection task, and increases in theta-gamma PAC have been linked with improvements in working memory in older adults [64, 65]. Although these studies are informative, none included tasks with a spatial component, and the n-back tasks that were used placed relatively low demands on sequence memory (e.g., 2 item sequences). While these collective findings of altered frontal activity for both spatial and temporal working memory can be related to prominent theories about frontal lobe deterioration and/or compensation with aging [66, 67], it remains unclear how changes that occur with healthy aging are reflected in the oscillatory dynamics during VSWM encoding and recall for sequences.

## 1.5 Hypotheses

The primary aims of the present study were (1) to determine whether oscillatory dynamics in the Theta and Gamma ranges would reflect item-level sequence *encoding* during a computerized spatial span task, (2) to determine whether item-level sequence *recall* is also reflected in these neural oscillations, and (3) to determine the nature of potential changes to these processes in healthy cognitive aging. In light of the well-established changes in VSWM and sequence memory with aging, behavioral and EEG measures were collected from both younger and older adults in order to determine whether there would be evidence for altered brain function during encoding and/or retrieval in this population. Behaviorally, we expected older adults to evidence a reduced VSWM capacity as measured by the average length of correctly recalled sequences. In line with available literature suggesting that gamma band power increases with memory load [17], we expected gamma power at encoding would increase monotonically with subsequent items presented in a spatial sequence and that these item-level changes in gamma power would be diminished in older adults. We likewise expected that theta power would increase monotonically with subsequent items at encoding and that theta power would be diminished in older adults. Since previous work suggests that theta-gamma PAC decreases with subsequent items presented in a sequence [14], we expected that PAC would be observed to decrease monotonically with serial position during encoding. Furthermore, based on literature suggesting that theta-gamma PAC is reduced in older adults [64, 65], we also expected that younger adults would also show greater PAC magnitude relative to older adults.

Given that a vast majority of the extant literature has focused on the comparison of correctly and incorrectly recalled memories and has emphasized the retention period of visuo-spatial sequence memory rather than event markers that could be used to time-lock item recall, our expectations with respect to VSWM recall were far more exploratory. To this end, we used participant's responses (as they recalled items in the sequence) with the assumption that the mouse click reflecting the participant's recall of item N could be used as a time-locking event marker of the initiation of the recall of item N+1. Based on the available literature [13, 45], we expected that oscillatory power in the theta range would decrease monotonically as each item in the sequence was recalled. Likewise, given the possibility that working memory load is decreased as the sequence is recalled, we expected a similar pattern in the gamma band. Given the aforementioned finding that PAC has been demonstrated to be reduced in magnitude with the increased demands associated with the subsequent presentation of items in a sequence and the possibility that these working memory demands are monotonically decreased as the sequence is recalled, we expected that the strength of theta-gamma PAC may likewise increase monotonically with the recall of items in the sequence.

## 2. Method

### 2.1 Participants

Power analysis indicated a target sample size of 46 in order to detect a medium effect size ($f$ = 0.25) in the between-within subject interaction, assuming power equal to 0.9, alpha equal to 0.05, and a correlation between repeated measures of 0.5. After obtaining written informed consent, 79 (31 older) adults were recruited to participate in the present study. Data collection occurred between June and December of 2019. Four younger adult participants were excluded after failing to complete the experiment. One participant was excluded due to the loss of their demographic data. Five (three older adult) participants were excluded from data analysis based on exclusion criteria that included a self-reported history of treatment for epilepsy, Alzheimer's disease (or other memory disorder), schizophrenia, PTSD, Obsessive Compulsive

Disorder, and/or head injury involving a loss of consciousness. Finally, eight older adult participants and four younger adults were excluded because they achieved fewer than 10 correct trials or fewer than 10 usable trials remained (after data preprocessing) in any of the trial bins used for behavioral and/or oscillatory analyses at encoding. Thus, the final sample consisted of 59 participants. The majority of the sample was right handed ($n$ = 49). Older adults aged 65 years or older ($n$ = 20; 60.0% female) were recruited using fliers posted at the town's local library, recreation center, and assisted living facilities as well as online announcements to local Facebook groups. The average age of the older participants was 70.0 years ($SD$ = 3.81). Older adult participants were not reimbursed but were offered a token gift for their participation. All but one of the older adults in the sample had attained at least an Associates degree at the time of their participation in this study and 35% ($n$ = 7; Masters = 6, Doctorate = 1) had attained a graduate degree. Younger adults ($n$ = 39; 67.5% female) were recruited from a small Southeastern public liberal arts university and were awarded course credit for their participation in the study. Younger adults were an average age of 19.9 ($SD$ = 4.47) years old.

## 2.2 Procedure

After providing informed consent, participants were assigned a unique identification number that was linked with all of the data collected. This number was not linked with the participant's identifying information at any time. All participants then completed a short demographic questionnaire and older adult participants also completed the 30-item Montreal Cognitive Assessment [68], followed by the experimental task. EEG recordings were made while participants were seated in a dimly lit and electromagnetically shielded room, positioned approximately 60 cm away from the computer monitor. All participants were presented with self-paced, written instructions and provided an opportunity for clarification before the experiment began. This protocol was completed in about one hour for each participant.

## 2.3 Behavioral methods

**2.3.1 Neuropsychological measures.** Cognitive functioning was assessed in older adults using the 30-item Montreal Cognitive Assessment (MoCA) [68]. MoCA scores indicated the present sample included a comparatively healthy group of older adult participants with an average score of 27.7 ($SD$ = 1.81). Only one of the older adult participants scored below (24) the recommended optimal cutoff score of 25 [68], which is sometimes used for the detection of mild cognitive impairment (MCI). However, this participant was included in the analysis because they did not report being treated for any form of cognitive impairment and because exclusion of this participant did not have any qualitative impact on the pattern of statistical results in the analysis.

**2.3.2 Experimental stimuli.** All experimental stimuli were presented using PsychoPy2 [69]. Each trial of the memory task began with the text, "Get Ready", which remained on-screen for a duration of 1 second. After a 1 second interstimulus interval (ISI), participants were presented with a white 3x3 grid of open squares against a black background (see Fig 1). After a 1 second delay, a random series of squares would then illuminate (i.e., filled white) one at a time until the number of squares in the sequence reached the participant's estimated memory "span" (minimum 2, maximum 9). Squares were illuminated for a random duration of 750 ms to 1250 ms (uniform distribution) with an inter-stimulus interval between 750 ms to 1250 ms (uniform distribution). When the sequence was complete, the screen was cleared of all stimuli (including the grid) for a period of 2.5 seconds. After this 2.5 second maintenance delay, the original white grid reappeared along with the text, "Recall the sequence" and a button labeled "Done". Participants were asked to recall the sequence by clicking inside the squares of

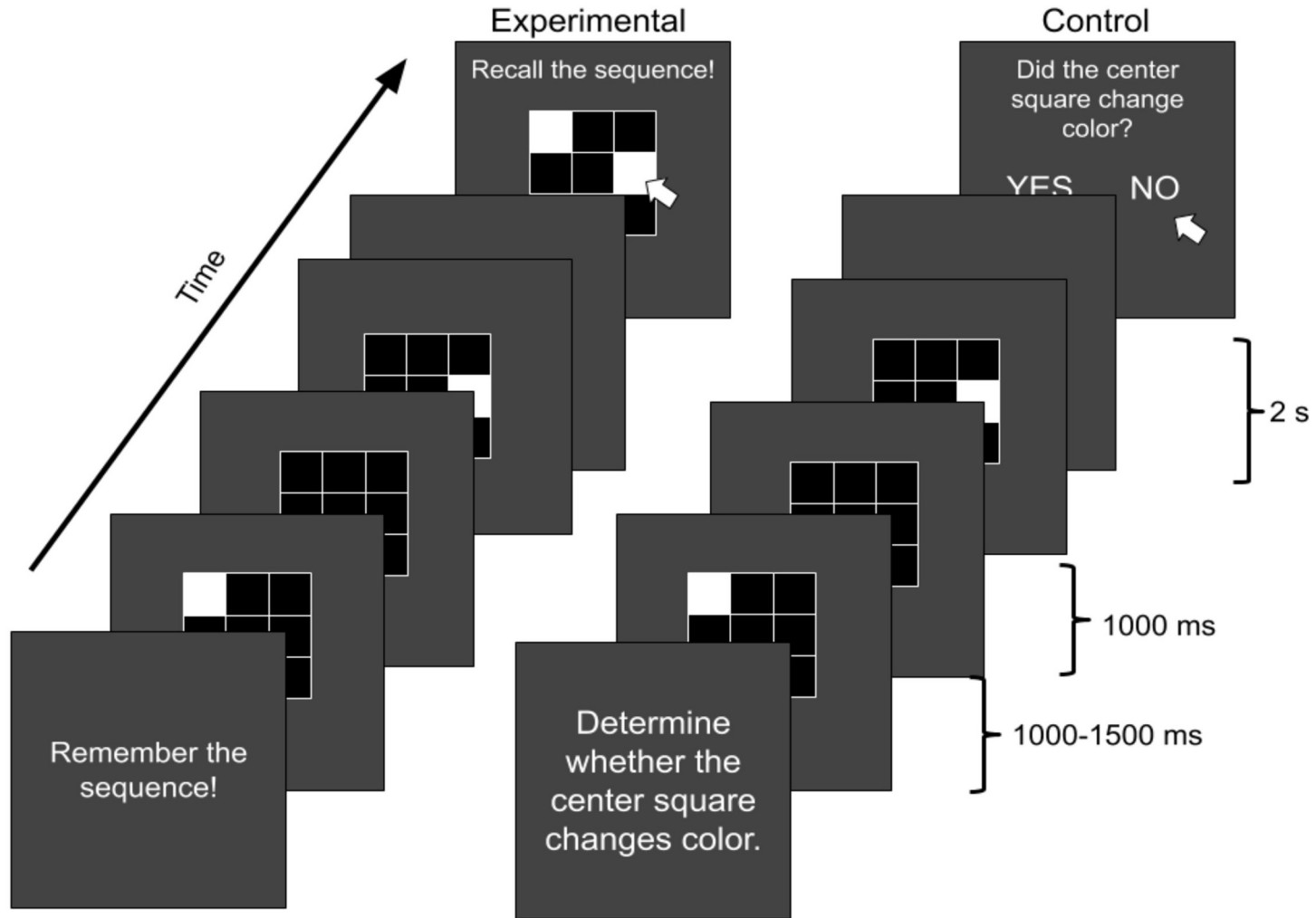

**Fig 1. Schematic of the spatial working memory and control tasks.**

the grid one at a time and to click the "Done" button when finished recalling the sequence. Squares were illuminated with each click by the participants and would remain illuminated during the recall period. Corrective feedback was presented for a period of 1 second when participants clicked the "Done" button. The word "Correct" was displayed if the participant correctly recalled both the square's locations and their sequential order. The word "Almost" was used if the participants correctly recalled the square's locations, but incorrectly recalled the sequential order. Finally, the word "Incorrect" was displayed if participants incorrectly recalled both the square's locations and their sequential order.

Memory span, which determined the length of each sequence, began at 4 on the first memory trial for all participants, but was adaptively re-estimated after each memory trial. Each time participants correctly recalled a sequence, their estimated span was increased by one. If participants correctly recalled the locations, but not the order of a sequence, their estimated span remained unchanged. Finally, if participants incorrectly recalled both the locations and order of a sequence, their estimated span was decreased by one.

Blocks of experimental trials were interleaved with a control task. The control task was identical to the experimental task with the following exception. Rather than being asked to

recall the sequence of squares, participants were asked only to determine whether or not the center square was illuminated at any point during the sequence. Following the presentation of a random sequence of squares, participants were presented with the text, "Did the center square change color?" in the center of the monitor and two buttons, arranged centrally below the text, that were labeled "Yes" and "No". The experiment consisted of four blocks of three control trials followed by ten experimental trials. Task instructions were presented to participants prior to each set of control and experimental trials within a block.

**2.3.3 Measures of behavior.** The experimental task yielded two primary behavioral measures. The first is the memory span achieved over the course of the experiment, which is analogous to Corsi Span measured in prior research [10]. The second measure was the reaction times (RT) during sequence recall. The first RT was the measure of the time between the appearance of white grid and the first time participants clicked in one of the squares. All subsequent measures of RT were calculated with respect to the prior response. For example, the recall RT for the second item in a sequence was calculated as the time between the first RT and the second click inside one of the squares. Accuracy was also recorded as the percent of Correct, Almost, and Incorrect responses over the course of the experiment.

## 2.4 EEG methods

**2.4.1 EEG recording and processing.** Electrophysiological data were recorded at a sampling rate of 2000 Hz using a high-impedance DBPA-1 Sensorium bio-amplifier (Sensorium Inc., Charlotte, VT) with an analog high-pass filter of 0.01 Hz. All recordings were made using an extended 10/20 cap system with 65 Ag-AgC1 sintered electrodes with the reference electrode placed at the tip of the nose and the ground (i.e., common) electrode placed at the center of the forehead.

EEG data were pre-processed using EEGLAB [70] and ERPLAB [71], and were analyzed using Brainstorm [72] in MATLAB (The Mathworks, Matick, MA). Raw data were resampled at 1000 Hz. An IIR Butterworth band-pass filter of 1–75 Hz (12dB/oct) was applied for the purpose of ocular artifact correction. The data were then visually inspected for extreme artifacts. Individual channels containing excessive artifact were noted and epochs containing excessive artifact across channels were removed from the continuous data. The data, excluding bad channels and segments, were then submitted to Independent Components Analysis (ICA) [73]. The ICA components were then automatically classified using the ICLabel plugin for EEGlab [74], and components labeled as eye movements and muscle activity above 80% probability were rejected [75]. The retained independent components were then back-projected to the raw, downsampled (1000 Hz) data (i.e., 65-channel electrode space). Bad channels identified in the previous step for participants were then interpolated for 14 of the participants using a spherical spline. Of those 14 with identified bad channels, the minimum number of channels interpolated was one ($N = 7$), the maximum number was four ($N = 4$), and two channels were interpolated for the remaining three participants.

The continuous recordings were segmented between -1000 and 2000 ms with respect to stimulus onset in the case of the encoding analysis and with respect to response execution (i.e., mouse click) in the case of the recall analysis. Segments were baseline corrected between -200 and 0 ms for visualization of the event-related potential (ERP) waveforms. Bad channels within each data segment were identified using a maximum-minimum voltage threshold of 200 μV. Finally, segments with more than 7 (10%) channels marked as bad were excluded from the analysis, resulting in the removal of an average of 1.0 (SD = 2.2) trials per subject across all conditions. The number of trials rejected did vary significantly across the three encoding conditions, $F(2,116) = 5.44$, $p < 0.01$, but did not significantly differ across the three recall

conditions, $F(2, 116) = 2.23$, $p = 0.11$. The potential confound introduced by varying the number of trials across sequence positions was mitigated by the systematic selection of an equal number of trials (described below).

Finally, in order to avoid the potential for confounding analyses of the position of stimuli in the sequence with the number of correct trials included in the analysis, sequence positions were aggregated into consecutive pairs 1–2, 3–4, and 5–6 [14] and the minimum number of correct encoding trials in each paired sequence group was used to determine the number of trials to be randomly selected from all of the three sequence-groups and control trials for each participant. For example, if we recorded 100, 54, 48, and 39 correct, usable trials for the Control, $Stim_{1-2}$, $Stim_{3-4}$, and $Stim_{5-6}$ events, then 39 trials would be randomly (uniform and without replacement) selected from each of the four stimulus conditions for that participant. When paired sequence groups were constructed for the recall analysis, there were three additional participants (2 older adults) in which the number of trials fell below 10 in one or more of the trial bins. These participants were only excluded from the recall analysis because excluding them from the behavioral and/or oscillatory analyses at encoding did not change the qualitative pattern of statistical results.

**2.4.2 Measures of ERPs and oscillatory dynamics.** In order to quantify the oscillatory data recorded from all sensors and, thus, to most accurately and comprehensively capture the spatial and temporal dynamics of both induced and evoked brain responses, spatial principal components analysis (PCA) was conducted using the ERP PCA Toolkit [76] in Matlab (The Mathworks, Matick, MA). The present application of PCA was modeled after prior research [77] in which PCA components were used as spatial filters to characterize oscillatory dynamics in multi-channel EEG data. Unlike prior research, which uses the inter-channel covariance in the time-domain ERPs and therefore emphasizes low-frequency, evoked responses in the derivation of spatial components, the present analyses uses the inter-channel covariance in the time-frequency domain, namely the event-related spectral perturbations (ERSP) [78], which measures the combined induced and evoked oscillatory activity.

First, a Morlet wavelet-based time-frequency analysis was conducted over 40 logarithmically spaced frequencies between 3 and 75 Hz over the interval between -1000 ms and 2000 ms with a mother wavelet characterized by a time resolution of 3 seconds at a central frequency of 1 Hz. Thus, the full-width at half-maximum(FWHM) ranged from 1 to 0.04 seconds with increasing wavelet peak frequency. This resulted in a spectral FWHM range of 0.88 Hz to 22.06 Hz. The ERSP values were then computed with respect to a baseline interval of -500 to -50 ms and averaged over trials for each participant, separately for each of the Control and aforementioned paired sequence position groups for the encoding stimuli (e.g., E1-2, E3-4 and E5-6) and responses (e.g., R1-2, R3-4, and R5-6). For both PCA analyses, ERSP dynamics specific to the mnemonic, sequence processing were isolated by subtracting the ERSP values obtained from the Control condition from each of the three paired sequence conditions at encoding and recall. For the encoding PCA analysis, the averaged ERSP data at 261 time points between -300 and 1000 ms were reorganized into a Times (261) * Frequencies (40) * Condition (3) * Subjects (59) X Channels (65) matrix. Principal components were then derived from the channel covariances in this 1,847,880 X 65 element matrix. For the recall PCA analysis, the averaged ERSP data between -300 and 1000 ms were similarly reorganized, into a Times (261) * Frequencies (40) * Condition (3) * Subjects (56) X Channels (65) matrix and principal components were then derived from the channel covariances in this 1,753,920 X 65 element matrix. A parallel test [79] was used to determine the number of components to retain in each of the encoding and recall PCAs and those retained components were submitted to promax rotation and Kaiser normalization [80].

Each of the principal components can be considered a "virtual electrode" and the component scores were used to reconstruct virtual ERSP spectrograms analogous to the virtual ERPs used in prior research [81–83]. A subset of ERSP time-courses were then created for each subject, condition, and principal component by averaging the reconstructed ERSP data over the theta (4–7 Hz), low-gamma (25–50 Hz), and high-gamma (50–75 Hz) frequency bands.

## 2.5 Statistical analysis

The ERSP time-courses were analyzed using a mass univariate analysis [84] in which each of the 261 time points was submitted to a 2 (Group: Young/Old) X 3 (Position: E1-2, E3-4, E5-6) mixed-measures ANOVA. In order to address concerns about the inflation of family-wise error, clusters of significant main effects and/or interactions in these time courses were identified using a cluster-based permutation test with a conservative cluster inclusion $p$-value of 0.025 and a minimum cluster length of three consecutive time-points. Statistically significant clusters were identified as those associated with less than a 0.05% probability (i.e., $p < 0.05$) of being observed under the null hypothesis distribution generated from 1,000 random permutations of the Group and Position factors [84], separately for each frequency band but across PCA components. Finally, in order to simplify exposition of the results, mean ERSP values were computed over time points within each significant cluster and these cluster means were submitted to a mixed-measures ANOVA corresponding with that used during the mass univariate analysis. Greenhouse-Geisser correction was used in any/all cases of violation of the sphericity assumption at all stages of the analysis. Planned, pairwise comparisons were limited to those between serial positions *within* each age group.

## 2.6 Phase amplitude coupling

PAC was computed in Brainstorm by creating a PAC comodulogram with frequencies of 2–8 Hz for phase (x-axis) and 25–75 Hz (y-axis) for amplitude. The PAC algorithm in Brainstorm uses the mean vector length method of determining a "direct PAC" measure [85]. PAC was estimated over the interval between 0–1000 ms following stimulus onset for control and sequence encoding trials or following responses at sequence recall. The PAC comodulogram for control trials was subtracted from each of the comodulograms for the paired stimulus sequences at encoding (E1-2, E3-4, and E5-6) and recall (R1-2, R3-4, and R5-6) for each participant. The resulting PAC measures were then averaged across frequencies 3–7 Hz for phase and 25–50 Hz for amplitude in order to obtain a single estimate of theta-gamma PAC in the low gamma band at each channel. Similarly, PAC measures were averaged across frequencies 3–7 Hz for phase and 50–75 Hz for amplitude in the comodulogram in order to obtain a single estimate of theta-gamma PAC in the high gamma band at each channel. Only mean PAC at channel Fz was used for statistical analysis.

## 3. Results

### 3.1 Behavioral

**3.1.1 Memory span.** An independent samples t-test showed that memory span was significantly longer in the younger (M = 7.43, SD = 0.52) compared with the older adult (M = 6.45, SD = 0.65) group, $t(57) = 6.32$, $p<0.001$, $d = 1.74$.

**3.1.2 Accuracy.** Because the task was designed to keep participants at their estimated memory spans, there was no significant difference between age groups in the percentage of correct responses. However, younger adults committed significantly more "Almost" errors (i.e., correct location but incorrect order) compared with older adults, $t(57) = 2.76$, $p = 0.008$,

$d = 0.76$, while older adults committed significantly more "Incorrect" (i.e., incorrect location) errors responses compared to younger adults, $t(57) = 4.57$, $p < 0.001$, $d = 1.26$ (see Fig 2).

**3.1.3 Reaction time.** There were main effects of Position, $F(4,228) = 22.96$, $p < 0.001$, $\eta^2_p = 0.29$, and Group, $F(1,57) = 11.3$, $p = 0.001$, $\eta^2_p = 0.17$, on reaction time, but this interaction was not statistically significant (see Fig 2). The main effect of Group indicated that younger adults tended to respond faster (M = 880ms, SD = 25ms) during the recall of each spatial

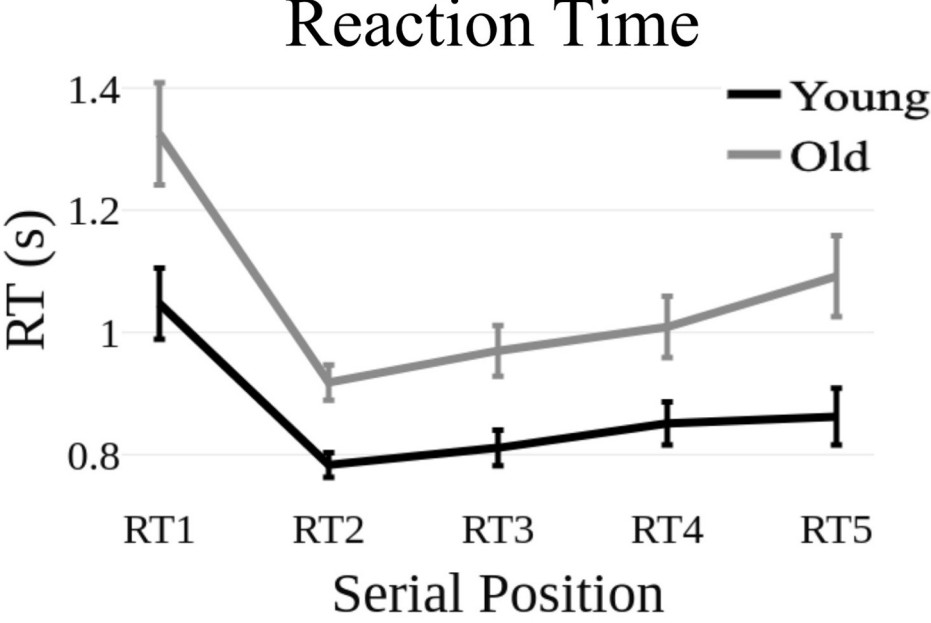

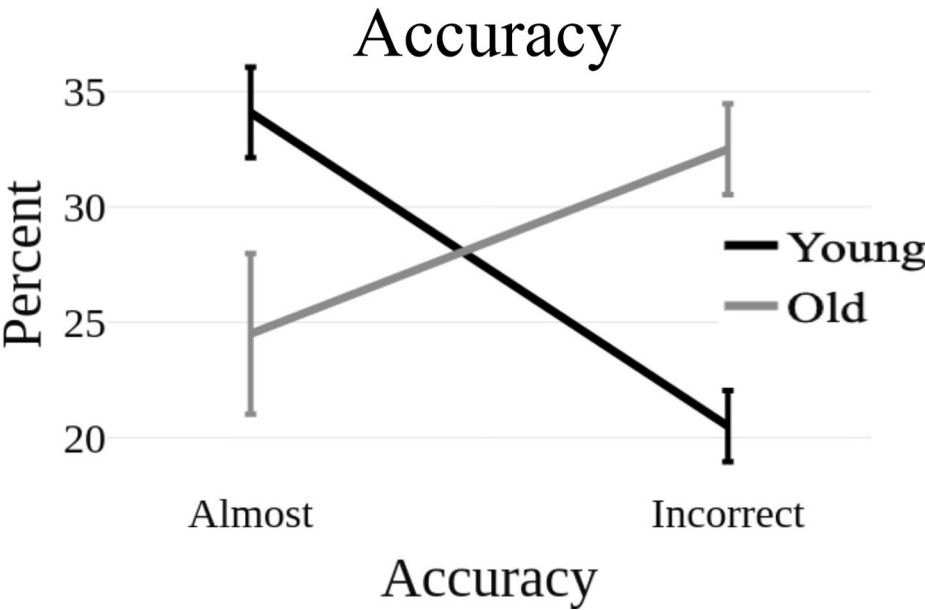

**Fig 2. TOP: Reaction times (click$_N$-click$_{N-1}$) at serial positions 1–5.** BOTTOM: Error rates for each of the two error types during sequence recall.

position compared with older adults (M = 1024ms, SD = 35ms). The main effect of Position was characterized by much longer reaction times for the first position in the recall of the sequence (RT1) compared with the second (RT2), $t(57)$ = 6.97, $p<0.001$, third (RT3), $t(57)$ = 5.61, $p<0.001$, fourth (RT4), $t(57)$ = 4.80, $p<0.001$, and fifth (RT5), $t(57)$ = 4.15, $p$ = 0.001 positions. Mean reaction times increased monotonically from RT2 to RT5 but the only other significant difference was between RT2 and RT5, $t(57)$ = 3.48, $p$ = 0.008. (see Fig 2).

## 3.2 EEG

**3.2.1 Encoding.** The raw, unfiltered grand-averaged ERP waveforms time-locked to the onset of task and control stimuli are illustrated in Fig 3.

**Component 1**. The first spatial component accounted for 33.8% of the variance in the Promax-rotated solution and was most strongly distributed over midline frontal sites (see Fig 4). There was a significant cluster in the high gamma band, but no other clusters reached statistical significance.

*High Gamma*. There was a main effect of Position in the high gamma cluster between 707–947 ms, $F(2,114)$ = 22.30, $p<0.001$, $\eta^2_p$ = 0.28. Gamma power increased monotonically across the three aggregated positions, but this increase was only statistically significantly in the comparison of encoding positions E1-2 compared with positions E3-4, $t(57)$ = 4.44, $p<0.001$, and positions E5-6, $t(57)$ = 5.66, $p<0.001$.

**Component 2**. The second spatial component accounted for 31.4% of the variance in the Promax-rotated solution and was most strongly distributed over midline posterior sites (see Fig 4). There was a significant cluster in the high gamma band, but no other clusters reached statistical significance.

*High Gamma*. There was a main effect of Position in the high gamma cluster at 782–947 ms, $F(2,114)$ = 20.58, $p<0.001$, $\eta^2_p$ = 0.27. Gamma power increased monotonically with the sequence positions, but these increases were only statistically significant in the comparisons of positions E1-2 with positions 3–4, $t(57)$ = -4.70, $p<0.001$, and positions E5-6, $t(57)$ = 5.54, $p<0.001$.

**Components 3–4**. Spatial components 3 and 4 accounted for 17.8% and 4.2% of the variance in the Promax-rotated solution, respectively. The cluster-based permutation test did not reveal any significant main effects or interactions for either components 3 or 4 (see Fig 4).

**Phase Amplitude Coupling**. There were no significant main effects of Position or Group or any significant interactions in measures of PAC in either the low or high gamma bands at encoding (see Fig 5A and 5C).

**3.2.2 Recall.** The raw, unfiltered grand-averaged ERP waveforms time-locked to responses at each paired sequence position during recall are illustrated in Fig 6.

**Component 1**. The first spatial component accounted for 34.5% of the variance in the Promax-rotated solution and was most strongly distributed over midline frontal sites (see Fig 7). There was a significant cluster in the Theta band, but no significant clusters in either the high or low gamma bands.

*Theta*. There was a significant main effect of Group in the theta cluster at 77–532 ms, $F(1,54)$ = 9.68, $p$ = 0.003, $\eta^2_p$ = 0.15, indicating increased theta power during recall in older compared with younger adults.

**Component 2**. The second spatial component accounted for 30.8% of the variance in the Promax-rotated solution and was most strongly distributed over midline posterior sites (see Fig 7). There were significant clusters in both the theta and low gamma ranges.

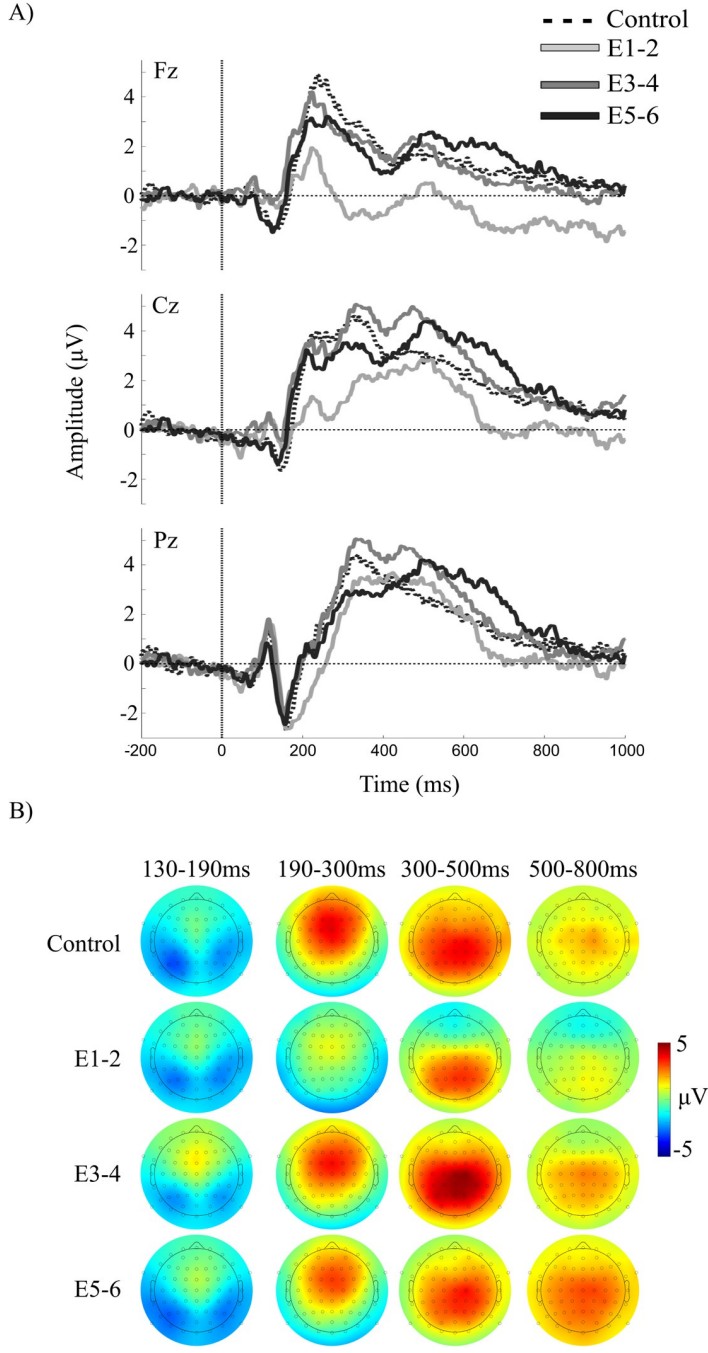

**Fig 3. Raw, grand-averaged ERPs at select midline electrode sites, time-locked to stimulus onset during the spatial memory and control tasks.** ERPs are presented separately for each of the paired groupings of sequence position during encoding, E1-2, E3-4, and E-5-6. Topographical maps illustrate the voltage distribution across all channels at each of four ERP components identified in the waveforms.

***Theta***. There was a significant interaction between Group and Position in the theta cluster at 377–592 ms, $F(2,108) = 4.25$, $p = 0.017$, $\eta^2_p = 0.07$. However, none of the pairwise comparisons were statistically significant (see Fig 7).

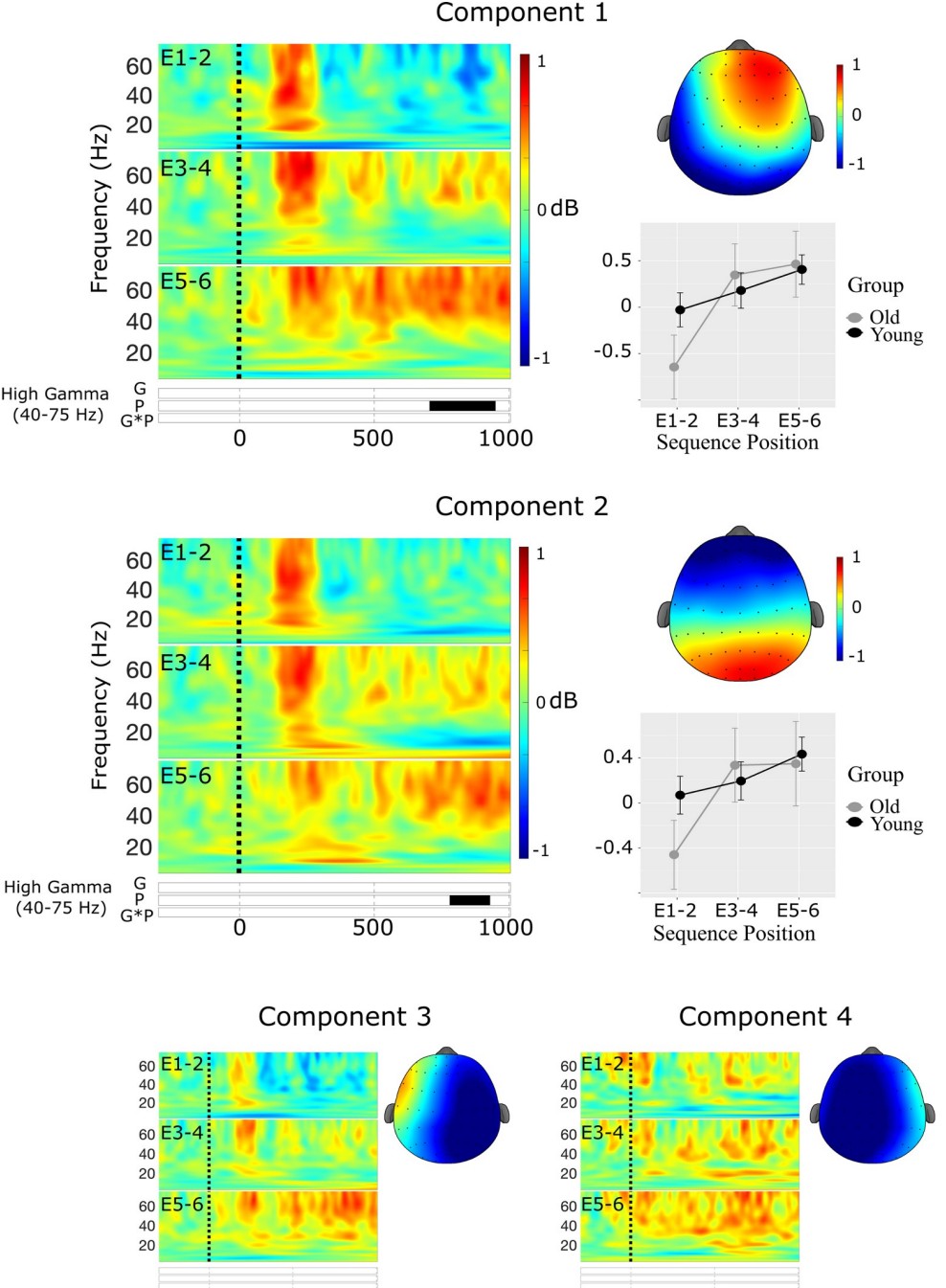

**Fig 4. Results of the spatial PCA and cluster-based permutation tests of ERSP at encoding.** Raster plots illustrate statistically significant epochs for the effects of Group (G), Position (P), and the interaction (GxP). Line charts illustrate the mean ERSP within each cluster across the paired encoding positions E1-2, E3-4, and E5-6. Note that the color scales are consistent across.

*Low Gamma*. There was a main effect of Group in the low gamma cluster at 27–152 ms $F$ (1,54) = 11.5, $p < 0.001$, $\eta^2_p = 0.18$, indicating increased low gamma power during recall in older compared with younger adults (see Fig 7).

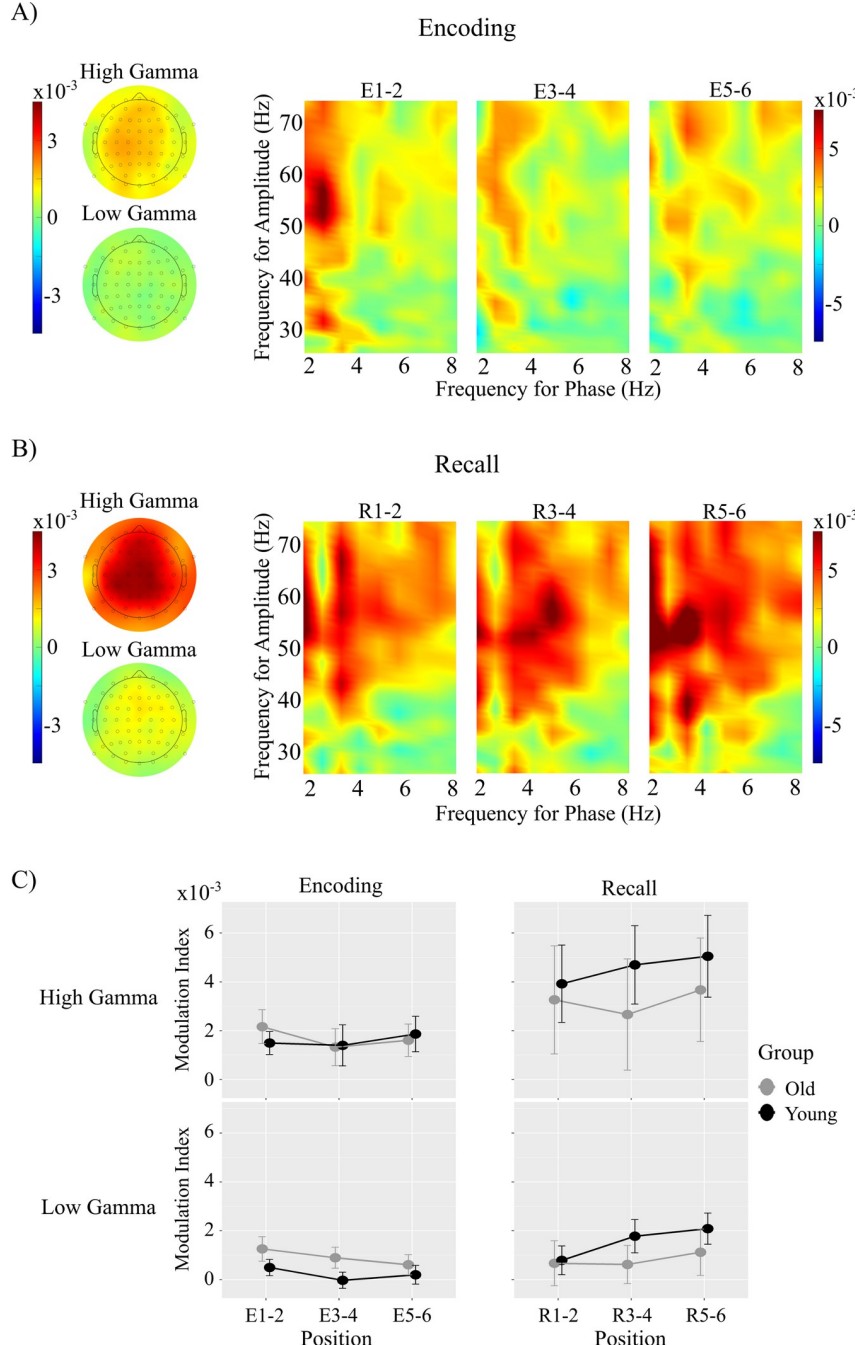

**Fig 5.** Results of the PAC analysis at (A) sequence encoding and (B) sequence recall. Topographical maps illustrate mean PAC at each channel. Comodulograms indicate PAC at channel Fz. (C) Line charts illustrating mean PAC values in the high and low gamma ranges at each of the paired serial positions at encoding (left) and recall (right).

**Component 3**. The third spatial components accounted for 13.4% of the variance in the Promax-rotated solution and was most strongly distributed over left lateral electrode sites. There were no significant clusters in any frequency band in component 3 (see Fig 8).

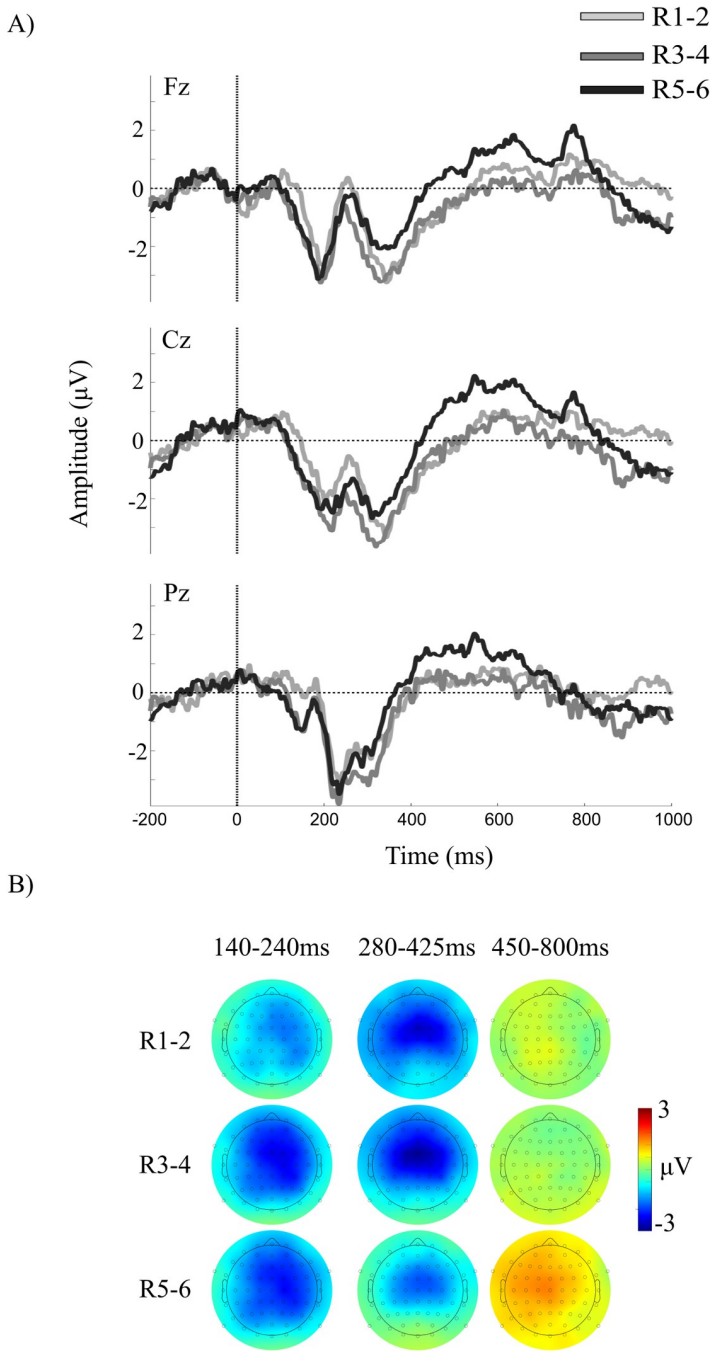

**Fig 6. Grand-averaged ERPs at select midline electrode sites, time-locked to responses at each paired sequence position R1-2, R3-4, and R5-6 during recall.** Topographical maps illustrate the voltage distribution across all channels at each of three ERP components identified in the waveforms.

**Component 4**. The fourth spatial component accounted for 8.9% of the variance in the Promax-rotated solution and was most strongly distributed over right lateral electrode sites (see Fig 8). There was a significant cluster in the theta band in component 4.

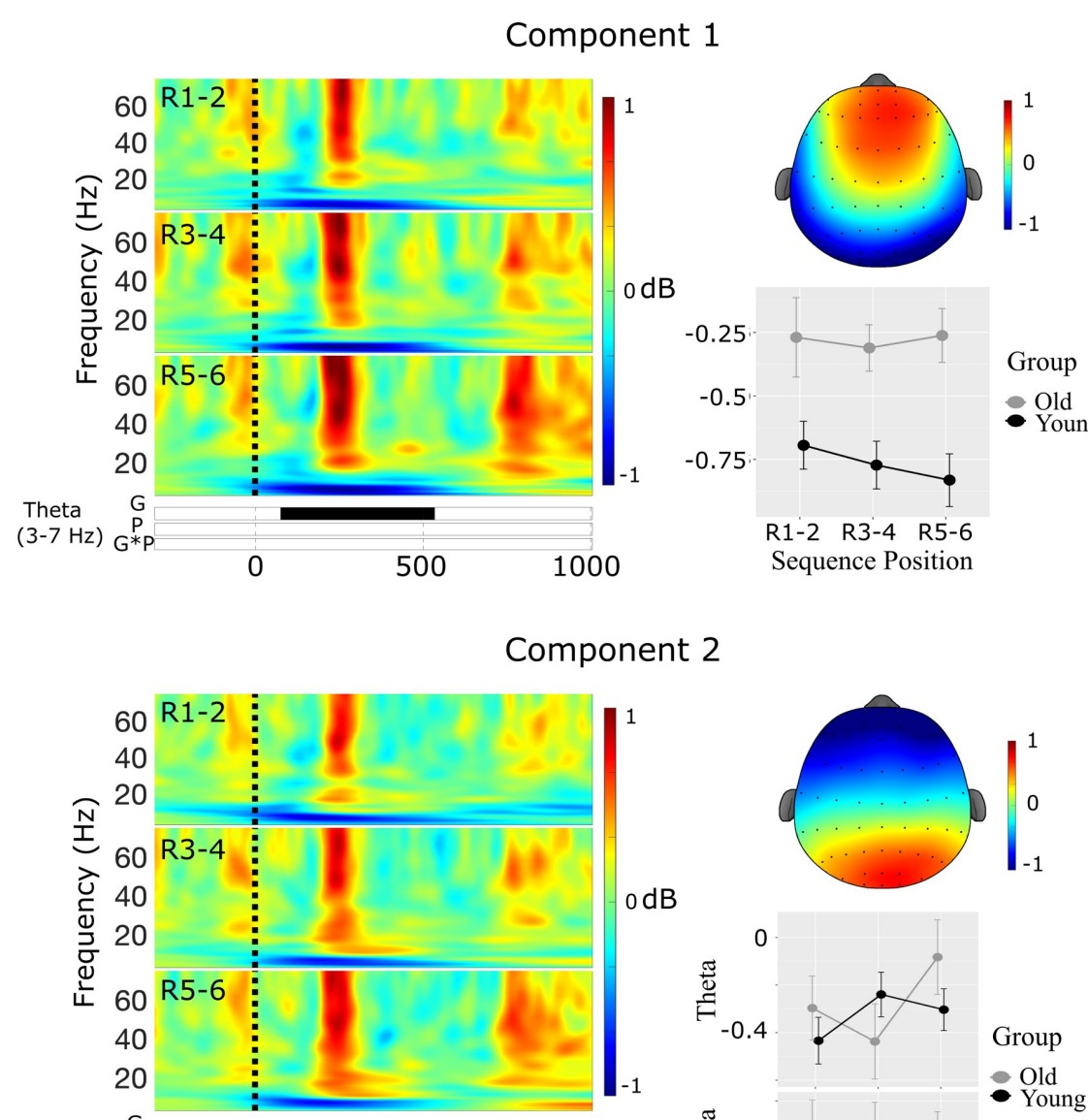

**Fig 7.** Results of the spatial PCA and cluster-based permutation tests for spatial components 1 (top) & 2 (bottom) at sequence recall. Raster plots illustrate statistically significant epochs for the effects of Group (G), Position (P), and the interaction (GxP). Line charts illustrate the mean cluster power at each of the paired sequence positions R1-2, R3-4, and R5-6.

*Theta*. There was a main effect of Group in the theta cluster at 667–1000 ms $F(1,54) = 6.87$, $p = 0.011$, $\eta^2_p = 0.11$, indicating an increase in theta power in younger compared with older adults (see Fig 8).

**Phase Amplitude Coupling.** *Low Gamma*. There was a significant main effect of Position for mean PAC in the low gamma band, $F(2,108) = 4.87$, $p = 0.009$, $\eta^2_p = 0.08$. Mean PAC

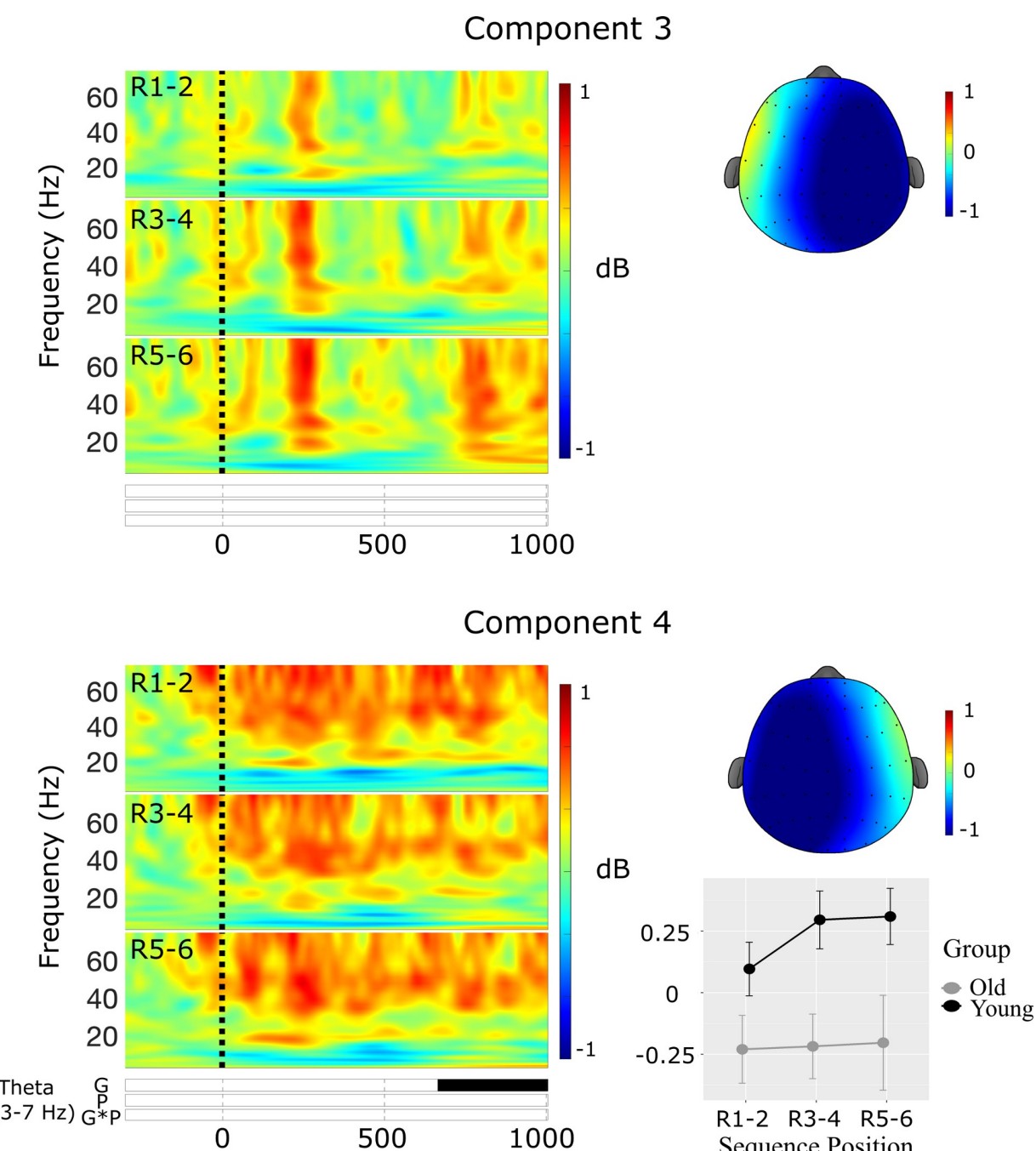

**Fig 8.** Results of the spatial PCA and cluster-based permutation tests for spatial components 3 (top) & 4 (bottom) at sequence recall. Raster plots illustrate statistically significant epochs for the effects of Group (G), Position (P), and the interaction (GxP). Line charts illustrate the mean cluster power at each of the paired sequence positions R1-2, R3-4, and R5-6.

increased monotonically with serial position, however the pairwise increase was only statistically significant between positions R1-2 and R5-6, $t(54) = 2.86$, $p = 0.016$ (see Fig 5). Neither the main effect of Group nor the interaction between Group and Position reached statistical significance.

*High Gamma*. There was also a significant main effect of Position for mean PAC in the high gamma band, $F(2,108) = 3.60$, $p = 0.031$, $\eta^2_p = 0.06$. Mean PAC increased monotonically with serial position, however the pairwise increase was only marginally statistically significant between positions R1-2 and R5-6, $t(54) = 2.29$, $p = 0.065$ (see Fig 5). Neither the main effect of Group nor the interaction between Group and Position reached statistical significance.

## 4. Discussion

The primary aims of the present study were (1) to determine whether oscillatory dynamics in the theta and gamma ranges would reflect item-level encoding during a computerized spatial span task and (2) to determine whether item-level sequence recall is also reflected in these oscillatory dynamics. A secondary aim of this study was to determine whether there would be evidence for differences in these item-level oscillatory dynamics in the theta and gamma frequency ranges during encoding and/or recall in healthy older adults. Taken together, the results support a growing body of literature suggesting the critical functional role of gamma and theta band oscillations in both the encoding and retrieval of content in working memory. Specifically, this research demonstrates that gamma-band oscillatory power at encoding and theta-gamma PAC at retrieval are each modulated at the item-level in VSWM for spatial sequences. Although no evidence was observed for alteration of these sequence-specific modulations of theta and gamma in healthy older adults, group differences in oscillatory power may reflect age-related changes and/or reflect compensatory responses.

### 4.1 Behavioral findings

As hypothesized, the older adult group evidenced a significantly shorter average memory span and significantly longer recall RTs compared to younger adults. Notably, the pattern of RTs across serial positions in the recall sequence (see Fig 2) is qualitatively similar to previous findings in the literature, and also similar in the younger and older adult groups in this study. This suggests that, to the extent to which the pattern of RTs reflects retrieval strategies (e.g., chunking) which has been demonstrated in prior research [48], observed differences between the two groups may not be attributable to changes in retrieval strategy.

Regarding the types of recall errors committed by participants, older adults were more likely than younger adults to incorrectly recall one or more stimulus locations, whereas the majority of errors committed by younger adults involved the recall of correct stimulus locations, but in an erroneous sequence. The tendency for older adults to commit spatial location errors aligns with previous literature suggesting that healthy aging impairs VSWM [86], while temporal memory may remain relatively intact [87]. Age-related decline in spatial location memory performance may also be associated with older adults' decreased use of spatial organization (i.e, categorical and spatial relationships), as supported by recent work [88].

### 4.2 Encoding

Consistent with our hypothesis, we observed a significant main effect of sequence position in the high gamma band (50–75 Hz) over medial-frontal sites at 707–947 ms following stimulus onset with high gamma power increasing monotonically with successive positions in the sequence. This same pattern was also observed to be significant over midline posterior sites. This finding is entirely consistent with prior research demonstrating that oscillatory power in the gamma band increases with working memory load [17, 35, 89]. The fact that these sequence specific modulations were limited to oscillations in the gamma band is also in line with findings that load-based modulations of oscillatory power during encoding are limited to gamma-band frequencies [90]. Consistent with previous work indicating that gamma-band

oscillations in posterior and occipital visual cortices evidence load-dependent modulation for specific sensory information, including stimulus location [91], we interpret the monotonic increases in gamma power over posterior electrode sites to likely reflect load dependent activation of stimulus location specific processes in the dorsal visual stream.

Because the majority of the extant literature investigating load-dependent modulation of gamma-band oscillations focuses on the delay period of short-term and/or working memory tasks, conclusions are typically limited to associations between the number of items held in memory and sustained gamma power over the delay interval. The present results expand on these results by offering insight into the timing of that load dependent modulation with respect to stimulus encoding as well as the nature of the sequence encoding process itself. First, the present findings suggest that there are coincident increases in high gamma power over both fronto-central and posterior recording sites that follow stimulus presentation by approximately 700 ms. In light of the fact that prior research suggests that this gamma band activity represents the maintenance of items in working memory, this invites the possibility that the monotonic increases in high gamma power observed in this study represent the point at which the current stimulus in a sequence of stimuli is encoded (i.e., incorporated) into a VSWM system that is already maintaining all previous stimuli in the sequence. Furthermore, the data suggest that this encoding of the Nth item in a sequence involves the recruitment of activity in both frontal and parieto-occipital neural networks.

Interestingly, despite clear behavioral differences between older and younger adults with respect to memory span, reaction times, and error types, there were no group differences or interactions between Group and Position observed in the load dependent modulation of high gamma power at encoding. One possibility is that visuo-spatial sequence encoding and its supporting neural networks were intact in this sample of healthy older adults. It is important to note that the present sample of older adult participants is particularly well-educated (19 of the 20 older adults attained at least one post-secondary degree), which has been linked to greater cognitive reserve [92], better VSWM performance [93], and preserved brain network organization with age [94]. Another possibility is that there is degradation of the networks supporting VSWM encoding in healthy aging, which leads to the observed behavioral changes, but that strategies to compensate for those performance deficits (e.g., increased motivation, attention, effort, etc.) result in high gamma band modulation that is comparable to younger adults. While it is impossible to discriminate between these possibilities in this study, they remain important considerations for future research.

### 4.3 Recall

The results of this study also provide support for the notion that time-locked responses that are recorded during the recall of a visuospatial sequence can be used to index item-level differences in brain activity as the recall of a sequence unfolds. Specifically, the magnitude of theta-gamma PAC was observed to increase monotonically with sequence position in both the low and high gamma bands. This pattern of results is consistent with our hypothesis based on previous research demonstrating a decrease in PAC magnitude with the storage of subsequent items in working memory due to the widening of the distribution of gamma power over the phase of theta [14]. In the context of sequence recall, this finding is consistent with the interpretation that working memory load (i.e., the number of items retained) is reduced as the sequence is recalled, resulting in the observed monotonic increases in PAC magnitude across the paired sequence positions.

Additionally, and contrary to our hypothesis, a prolonged *increase* (77-532ms) in theta power over midline frontal sites was observed in older compared with younger adults.

Although our initial hypothesis was based, in part, on previous research demonstrating that theta oscillations may reflect cross-temporal associations during recall at the initial learning of a random sequence [13], that study also reported decreases in frontal theta power following responses during sequence recall for well-learned sequences. Thus, our results can be interpreted to reflect that frontal theta power during sequence recall is reduced when that recall is less demanding and/or effortful and the increased difficulty of recall for the older adults, as reflected in the behavioral data, was associated with an increase in theta power.

An interaction between Position and Group in the theta band and a main effect of Group in the low gamma band were also observed over midline posterior electrodes during recall. The interaction between Position and Group was associated with a relatively small effect size ($\eta^2_\text{p}$ = 0.07) and evidenced non-monotonic changes in theta power across paired sequence positions that were in opposing patterns across older and younger adults. Although non-monotonic patterns of reaction times during sequence recall have been associated with specific retrieval strategies [48] and this so-called "chunking" is thought to be enabled by hippocampal theta [95, 96], the fact that the pattern of theta power across sequence position does not mirror the pattern of reaction times in this study and that none of the paired comparisons were statistically significant warrants caution in the interpretation of this result. With respect to the Group differences in gamma power observed over posterior sites, we are reluctant to interpret the finding given the timing and duration of this effect (27-152ms) with respect to the recall response.

Finally, Group differences in theta power were observed during recall over the right lateral recording sites. The late timing of this effect (667-1000ms) with respect to the recall responses, which overlaps with subsequence responses on average (see Fig 2) makes us reluctant to provide an interpretation in terms of sequence recall. However, it is noteworthy that recent findings suggest that resting-state theta power over right middle temporal gyrus is negatively correlated with age, which may account for these non-mnemonic age-related differences [97].

## 4.4 Future directions and limitations

Future research will be required in order to explore the intriguing possibility that scalp-recorded oscillatory activity may reflect item level neural dynamics in the encoding and recall of VSWM sequences. While it is difficult to fully dissociate activity related to the preparation and execution of motor commands from the recall processes in the present design, our item-level, analytic procedure suggests that there are theta-gamma PAC dynamics that are related to sequence position during recall, albeit an exploratory technique. One potential limitation to this interpretation, however, is that there may be consistent differences in how the cursor movements were prepared and/or executed across the sequence (e.g., preparation/execution of singular responses, or preparation/execution of "chunks" of responses across the sequence). Future research may endeavor to further dissociate recall responses from the recall processes themselves as well as to consider the potential contributions of broad-spectrum oscillatory activity.

One potential caveat to the interpretation of the present findings related to gamma-band oscillatory dynamics is that some gamma-band activity related to microsaccadic eye movements may remain even after using ICA for the correction of ocular artifact [98, 99]. When comparing conditions in which there is a difference in the number and/or magnitude of microsaccades, this residual gamma-band activity may be misinterpreted as being due to cognitive as opposed to oculomotor functioning. However, in addition to using optimized filtering parameters [100] for the training data used to derive the ICA components for ocular artifact correction in the present study, the randomization of sequence positions across trials and participants in both the experimental and control conditions assuages concerns that

saccade magnitude, direction, and/or duration may have varied systematically across positions in the sequences used. It is also important to note that the present analysis includes only those positions in each sequence that were *correctly* remembered, assuaging concerns that previously reported differences in the number/frequency of saccadic eye movement varies between remembered and forgotten images [100].

It is noteworthy that despite demonstrating significant modulation of gamma-band oscillatory power at encoding and theta-gamma PAC at recall, there was minimal evidence for any differences between younger adults and older adults despite significant differences in VSWM task performance. This unexpected result invites the possibility that while both behavioral performance and measures of oscillatory dynamics during working memory tasks have been well documented in older adults, the integrity of those cognitive processes that are specific to the item level encoding and recall of stimuli in a visuospatial sequence remain preserved. Another consideration is that our sample was generally high-performing, and participants who did not achieve 10 correct trials were removed from analysis ($n = 12$, 8 older), which may limit generalizability to older adults with poorer cognitive performance. Of course, an important avenue for future research will be the investigation of these item-level oscillatory dynamics in individuals with indications of compromised cognitive functioning in order to better understand aging trajectories as they relate to VSWM for sequences.

## 4.5 Conclusions

In summary, this research demonstrates that VSWM sequence encoding is related to gamma band oscillatory dynamics. Specifically, high gamma power over fronto-central and midline posterior recording sites increased monotonically as items were added to a visuo-spatial sequence in both younger and older adults, potentially reflecting the categorical representation of items in the sequence. Finally, the recall of VSWM sequences was related to oscillatory dynamics in measures of theta-gamma PAC with monotonic increases in PAC as items in the sequence were recalled. In addition to the general conclusion that these oscillatory dynamics are differentially related to the encoding and recall of VSWM sequences respectively, this research suggests that they may be specific to the item-level encoding and recall processes that are preserved with healthy aging.

## Author Contributions

**Conceptualization:** Makenna B. McGill, Paul D. Kieffaber.

**Data curation:** Makenna B. McGill, Paul D. Kieffaber.

**Formal analysis:** Makenna B. McGill, Paul D. Kieffaber.

**Investigation:** Paul D. Kieffaber.

**Methodology:** Makenna B. McGill, Paul D. Kieffaber.

**Project administration:** Makenna B. McGill, Paul D. Kieffaber.

**Software:** Paul D. Kieffaber.

**Supervision:** Paul D. Kieffaber.

**Visualization:** Paul D. Kieffaber.

**Writing – original draft:** Makenna B. McGill, Paul D. Kieffaber.

**Writing – review & editing:** Makenna B. McGill, Paul D. Kieffaber.

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
