## [Decision Letter · Decision Letter 0]

18 Sep 2023

PONE-D-23-16543Event-related Theta and Gamma Band Oscillatory Dynamics During Visuo-spatial Sequence Memory in Younger and Older AdultsPLOS ONE

Dear Dr. Kieffaber,

Thank you for submitting your manuscript to PLOS ONE. After careful consideration, we feel that it has merit but does not fully meet PLOS ONE’s publication criteria as it currently stands. Therefore, we invite you to submit a revised version of the manuscript that addresses the points raised during the review process.

This is an exciting and well-conducted study. However, I invite the authors to substantially revise their manuscript by addressing all the comments raised by the two reviewers. This will improve the quality of the manuscript and can be reconsidered for acceptance.

We look forward to receiving your revised manuscript.

Kind regards,

Vilfredo De Pascalis

Academic Editor

PLOS ONE

Journal Requirements:

We will update your Data Availability statement on your behalf to reflect the information you provide.3. Please include your full ethics statement in the ‘Methods’ section of your manuscript file. In your statement, please include the full name of the IRB or ethics committee who approved or waived your study, as well as whether or not you obtained informed written or verbal consent. If consent was waived for your study, please include this information in your statement as well.

Additional Editor Comments:

This is an exciting and well-conducted study. However, I invite the authors to substantially revise their manuscript by addressing all the comments raised by the two reviewers. This will improve the quality of the manuscript and can be reconsidered for acceptance.

Reviewers' comments:

Reviewer's Responses to Questions

**Comments to the Author**

1. Is the manuscript technically sound, and do the data support the conclusions?

Reviewer #1: Yes

Reviewer #2: Partly

2. Has the statistical analysis been performed appropriately and rigorously? 

Reviewer #1: Yes

Reviewer #2: Yes

3. Have the authors made all data underlying the findings in their manuscript fully available?

Reviewer #1: No

Reviewer #2: No

4. Is the manuscript presented in an intelligible fashion and written in standard English?

Reviewer #1: Yes

Reviewer #2: Yes

5. Review Comments to the Author

Reviewer #1: The present report tries to understand the time-frequency response of the human EEG during Visuospatial working memory (VSWM) for sequences of stimuli by means of the loading scores of principal components computed from Event-related spectral perturbations (ERSP). The focus would be in the encoding and recall processes given that the retention period has been studied more frequently. Given the working memory relationship with the theta-gamma phase-amplitude coupling (PAC) process, this parameter is also analyzed. One of the assets of the present report corresponds to the comparison of the obtained EEG-derived parameters in healthy young and older adults. The results show that encoding is related to gamma band with amplitude increases during the stimuli sequence, being this dynamic preserved in healthy older adults. During recall, PAC increased with the order position of the presented stimuli in both

age groups. Gamma seems to be crucial for neural processing during encoding and recall phases of VSWM, in both age groups, suggesting that this neural mechanism is important for VSWM in healthy aging. The rationale of the presented experiment is well justified, as well as the recording and analytical methods. Only certain precisions for the selection of certain methods are needed to make more clear the report.

Introduction:

-Some recent TF analyses of WM would complement the present introduction for the phases of encoding and retention. The latter point is particularly important because is not easy to discriminate between the ending of encoding and the start of the maintenance period (https://pubmed.ncbi.nlm.nih.gov/36958141/).

-The different models accounting for the neural mechanisms related to WM are not commented on, at least the distinction between reverberatory mechanisms and synaptic facilitation should be introduced, and how the neural dynamics found in the present report bias the interpretation to one or the other.

-The introduction is rather long, I understand the difficulty to be more concise. Is just one appreciation.

The hypotheses are clearly expressed, however, a diagram with the different expected trends for ERSP, and PAC would be desirable. By the way, the term Event-related Spectral Perturbation should be more extensively used across the Ms.

Methods.

-It is not clear why a control condition is needed. What is being excluded by the subtraction of control from the experimental condition?

xxxx

-Was the 1-75 Hz filter also applied for computing ERPs?

- Please indicate the number of cycles of the morlet-wavelets.

- Please justify the use of Promax rotation instead of the more standard varimax.

.-If I well understood the analytic process, loading scores of the 4-dimensional matrix permitted to compute loading scores for each channel, frequency, subject, and time for each principal component considered as statistically significant. Anyway, please express it in a pipeline. There was a definition of neighbors for the cluster mass technique?

- When computing PAC, theta was computed for 2-8 Hz and then averaged between 3-7 Hz, please justify.

Results

- Was cluster mass aplied to ERPs?

- With respect to ERPs were they computed with the 1-75 Hz filter?

- The cluster statistics in Figures 4-7, does it indicates the electrodes in which there was statistical significance.

- Similar for figures 6 and 8 for the PAC

- ERPs of the recall phase look very different to those of the encoding. In fact the negativities could be error-related negativities? . In fact one is computed from stimuli as trigger, and the other with the responses. Anyway the presentation in present report of ERPs, given that are not fully analyzed, seems to be a source of distraction

Discussion

Basically for discussion it could be potentially reduced in amplitude.

Reviewer #2: The manuscript reports the result of a study of visual working memory in older and younger adults. The manuscript is generally well written, the experimental design is generally appropriate, and the analyses are generally rigorous. I do however, have some concerns that prevent me from recommending the manuscript for publication in its present form. It may be possible to address these concerns in a revision. Below I list these concerns in the order they appeared in the manuscript, not necessarily in the order of seriousness.

The introduction seemed quite long. About 2 pages longer than it needed to be to motivate the actual study. It read more like a review paper than the introduction to a short empirical report. I don't think this detracts from the rigor of the work, so I wouldn't insist on any changes, but I do think busy readers would appreciate a more concise introduction.

I found the paragraph on lines 64-74 hard to follow. It lays out the critical gap in the literature which the current study aims to address, but it is not clear. I would suggest revising it to make the gap crystal clear. For example, why does "emphasizing the retention period" "not lend itself to time-locked, sequence position-level analyses"? I think I can see what the authors mean, but it is not obvious.

There were a number of sentences that referenced findings but did not include citations. For example, the sentence starting on line 76 seems to me to require some citations.

The number of participants excluded seems quite high, especially among older adults. Overall, almost 25% of the original participants were excluded. For example, 8 older and 4 younger participants were excluded for not having enough usable trials. This high exclusion rate raises concerns about selection effects, especially given that it was disproportionally high for older adults. I worry that the worst preforming participants were removed thereby reducing generalizability.

Relatedly, were any efforts taken to ensure that the older and younger adult samples were matched on variables other than age? For example, given that the younger adults were college students, were the older adults college graduates?

I am concerned about the use of visual inspection for rejecting bad epochs: "The data were then visually inspected for extreme artifacts. Individual channels containing excessive artifact were noted and epochs containing

excessive artifact across channels were removed from the continuous data." My worry is that this introduces too much subjectivity. Most studies use some objective definition to define an epoch as bad such as voltage deviations above some number of standard deviations. At a minimum I'd like more information on who inspected the data: were they blind to the purposes of the experiment? Were they blind to the age group of the subject? Were they blind to the condition of the epochs? But really, I think a more rigorous and objective method of detecting bad data should be used.

Line 358: how many segments were excluded due to >10% bad channels?

What are the error bars on Figure 2?

Finally, the data for the manuscript are not easily available. The authors state “Data are available from the corresponding author upon request.” but this seems to violate the PLOS statement that “Stating ‘data available on request from the author’ is not sufficient. If your data are only available upon request, select ‘No’ for the first question and explain your exceptional situation in the text box.”

6. PLOS authors have the option to publish the peer review history of their article (what does this mean?). If published, this will include your full peer review and any attached files.

Reviewer #1: **Yes: **CARLOS M. GOMEZ

Reviewer #2: No

---

## [Author Response · Author response to Decision Letter 0]

3 Nov 2023

Reviewer #1: The present report tries to understand the time-frequency response of the human EEG during Visuospatial working memory (VSWM) for sequences of stimuli by means of the loading scores of principal components computed from Event-related spectral perturbations (ERSP). The focus would be in the encoding and recall processes given that the retention period has been studied more frequently. Given the working memory relationship with the theta-gamma phase-amplitude coupling (PAC) process, this parameter is also analyzed. One of the assets of the present report corresponds to the comparison of the obtained EEG-derived parameters in healthy young and older adults. The results show that encoding is related to gamma band with amplitude increases during the stimuli sequence, being this dynamic preserved in healthy older adults. During recall, PAC increased with the order position of the presented stimuli in both age groups. Gamma seems to be crucial for neural processing during encoding and recall phases of VSWM, in both age groups, suggesting that this neural mechanism is important for VSWM in healthy aging. The rationale of the presented experiment is well justified, as well as the recording and analytical methods. Only certain precisions for the selection of certain methods are needed to make more clear the report.

We thank the reviewer for their thoughtful comments and have made every effort to reduce the length of the manuscript while adding additional background requested. We have also clarified the manuscript in response to each of the points raised by the reviewers. The reviewer’s comments and our responses are itemized below.

Some recent TF analyses of WM would complement the present introduction for the phases of encoding and retention. The latter point is particularly important because is not easy to discriminate between the ending of encoding and the start of the maintenance period (https://pubmed.ncbi.nlm.nih.gov/36958141/).

We appreciate the reference provided by the reviewer and it is has been incorporated into our discussion of the background literature in section 1.1 of the introduction.

The different models accounting for the neural mechanisms related to WM are not commented on, at least the distinction between reverberatory mechanisms and synaptic facilitation should be introduced, and how the neural dynamics found in the present report bias the interpretation to one or the other.

We appreciate the reviewer’s comments and recognize that synaptic facilitation and reverberation are important concepts related to our understanding of short term memory. However, we understand the concepts to be most relevant in the context of cellular neuroscience research (e.g., single-cell recordings & cell cultures) and computational neuroscience research (e.g., mathematical models) related to short-term and/or working memory. Unfortunately, reviewing the background necessary to support a discussion of these concepts would add considerably to the length of the manuscript (that we have been asked by both reviewers to reduce). Moreover, although the concepts are critical to our understanding of the cellular mechanisms of short-term memory in the brain, it is unclear how the load-dependent modulations of scalp-recorded oscillatory activity that are the focus of the current manuscript could contribute to the contemporary debate about the role that each may play in the maintenance of short-term memory traces. Thus, we respectfully argue that the addition of this literature would only distract from the most relevant background literature and that our data could not be used to make any strong claims about the role of these processes in spatial sequence memory.

The introduction is rather long, I understand the difficulty to be more concise. Is just one appreciation.

We appreciate the reviewer’s concern and have made an effort to be more concise in the revised manuscript while also adding the information requested by the reviewers. For example, we removed the discussion of fMRI findings from the introduction section, as well as some methodological details of the cited EEG/MEG studies.

The hypotheses are clearly expressed, however, a diagram with the different expected trends for ERSP, and PAC would be desirable. By the way, the term Event-related Spectral Perturbation should be more extensively used across the Ms.

We thank the reviewer for their comments. We understand that the reviewer would prefer that we refer to ERSP and PAC rather than use the more generic “oscillatory dynamics”expression that is used frequently throughout the manuscript. We have made every effort to replace that expression with ERSP and PAC where appropriate in the revised manuscript. As the reviewer suggests, the hypotheses are clearly expressed. Given that the hypothesis is that sequence-dependent changes will be monotonic, a graphical illustration would only consist of a straight line connecting three points at an arbitrary angle. We respectfully argue that the simplicity of such a plot would not contribute any further clarity to the manuscript.

It is not clear why a control condition is needed. What is being excluded by the subtraction of control from the experimental condition?

Subtraction of the control condition permits the subtraction of any oscillatory activity related to sensory and perceptual mechanisms that are common to the presentation of the experimental stimulus, but are unrelated to mnemonic processing. The reviewer is correct that subtraction of the control condition does not change the outcome of the sequence position analysis that was central to this work, however, it does prevent those processes from driving the results of the PCA analysis. In both ERP and ERSP data, high-amplitude/power during the first few hundred milliseconds of the response following a stimulus is associated with strong correlations between channels, which can strongly influence the decomposition of the data by PCA. Thus, we subtracted the response in the control condition to maximize the extent to which both data-reduction (i.e., PCA) and data analysis were based on the mnemonic processing of interest. We have reworded the sentence on page 18, line 391-392 of the original manuscript to clarify the contribution of the control condition.

Was the 1-75 Hz filter also applied for computing ERPs?

No. We have added the word “raw” at all places in the revised manuscript where the ERPs are referenced. Please note that the ERPs are illustrated only in an effort to be transparent about the data in its raw form and provide some context for the interpretation of the ERSP dynamics.

Please indicate the number of cycles of the morlet-wavelets.

While some software determines the temporal and spectral resolution of the wavelet analysis using a “number of cycles” parameter, the software used here (Brainstorm) instead designs the wavelet analysis using parameters for the central frequency and a full width half maximum (FWHM) in seconds of the mother wavelet. It has been argued that this is the best way to describe the wavelet transform in scientific communication (see Cohen, 2019). We have added the following sentences to the revised manuscript in order to better clarify the temporal and spectral resolution of the wavelet decomposition in our analysis.

“Thus, the full-width at half-maximum(FWHM) ranged from 1 to 0.04 seconds with increasing wavelet peak frequency. This resulted in a spectral FWHM range of 0.88 Hz to 22.06 Hz.”

Cohen, M. X. (2019). A better way to define and describe Morlet wavelets for time-frequency analysis. NeuroImage, 199, 81-86.

Please justify the use of Promax rotation instead of the more standard varimax.

To the best of our knowledge, Promax is the preferred component rotation method for PCA used with EEG/ERP data because the orthogonality constraint of the varimax method is typically considered biologically implausible. We referenced Dien et al. (2005) in our description of the PCA analysis.

Dien, J., Beal, D. J., & Berg, P. (2005). Optimizing principal components analysis of event-related potentials: matrix type, factor loading weighting, extraction, and rotations. Clinical neurophysiology, 116(8), 1808-1825.

See also…

Dien, J., Khoe, W., & Mangun, G. R. (2007). Evaluation of PCA and ICA of simulated ERPs: Promax vs. Infomax rotations. Human brain mapping, 28(8), 742-763.

If I well understood the analytic process, loading scores of the 4-dimensional matrix permitted to compute loading scores for each channel, frequency, subject, and time for each principal component considered as statistically significant. Anyway, please express it in a pipeline.

We appreciate the reviewer’s comment, but are unclear how exactly the PCA analysis could be expressed as a “pipeline” and have not been able to find any examples of such in the literature. The spatial PCA analysis reduces the number of channels to a smaller number of “components”. Thus, if I’m understanding the reviewer’s comment correctly, the “loading scores” are displayed topographically. The data for each frequency, subject, and time were extracted for each of the spatial components and analyzed using the cluster mass statistic.

There was a definition of neighbors for the cluster mass technique?

Yes. The cluster definition is stated on page 19, line 416 of the manuscript.

“...minimum cluster length of three consecutive time-points”

When computing PAC, theta was computed for 2-8 Hz and then averaged between 3-7 Hz, please justify.

PAC was computed over a slightly wider frequency range for the purposes of display only. We have added clarification to the caption for figure 5, reiterating that the analyses used PAC values between 3-7Hz (phase) and 25-50Hz (low gamma amplitude) & 50-75Hz (high gamma amplitude). 

Was cluster mass applied to ERPs?

No. The ERPs are presented only to offer readers transparency to the raw data and to provide some context for interpretation of the ERSP dynamics, which will make it easier to integrate the findings with some of the ERP-based literature regarding working memory.

The cluster statistics in Figures 4-7, does it indicates the electrodes in which there was statistical significance.

The cluster statistics were all based on the results of the PCA analysis. There were not any statistical tests conducted on individual channels.

Similar for figures 6 and 8 for the PAC

With respect to figures 6 & 8, the cluster statistics were all based on the results of the PCA analysis. There were not any statistical tests conducted on individual channels. With respect to the PAC analysis, it is stated on page 20, line 440 that PAC values were only calculated at channel Fz. We have added a second indication in the caption for figure 5 (PAC analysis) that the data reflect values computed at channel Fz.

ERPs of the recall phase look very different to those of the encoding. In fact the negativities could be error-related negativities? . In fact one is computed from stimuli as trigger, and the other with the responses. Anyway the presentation in present report of ERPs, given that are not fully analyzed, seems to be a source of distraction.

We appreciate the reviewer’s concern that the ERP plots may distract readers, however, we believe that they, more importantly, provide some context for the interpretation of the ERSP dynamics (especially the timing of the effects) and also promote the integration of the current findings with prior and future work using traditional ERP methods. I know that, as consumer of this literature and as a reviewer of similar work using ERSP, I appreciate having insight to the raw data used to generate the spectrograms.

Basically for discussion it could be potentially reduced in amplitude.

As with the introduction, we have made every effort to be more concise while adding the information requested by the reviewers in the revised manuscript.

Reviewer #2: 

The introduction seemed quite long. About 2 pages longer than it needed to be to motivate the actual study. It read more like a review paper than the introduction to a short empirical report. I don't think this detracts from the rigor of the work, so I wouldn't insist on any changes, but I do think busy readers would appreciate a more concise introduction.

We have made an effort to be more concise while adding the information requested by the reviewers in the revised manuscript. For example, we removed the discussion of fMRI findings from the introduction section, as well as some methodological details of the cited EEG/MEG studies.

I found the paragraph on lines 64-74 hard to follow. It lays out the critical gap in the literature which the current study aims to address, but it is not clear. I would suggest revising it to make the gap crystal clear. For example, why does "emphasizing the retention period" "not lend itself to time-locked, sequence position-level analyses"? I think I can see what the authors mean, but it is not obvious.

We agree that this paragraph could benefit from further clarification and have revised it per the reviewer’s suggestion.

There were a number of sentences that referenced findings but did not include citations. For example, the sentence starting on line 76 seems to me to require some citations.

We appreciate and agree with the reviewer’s comment. Citations have been added for this sentence.

The number of participants excluded seems quite high, especially among older adults. Overall, almost 25% of the original participants were excluded. For example, 8 older and 4 younger participants were excluded for not having enough usable trials. This high exclusion rate raises concerns about selection effects, especially given that it was disproportionally high for older adults. I worry that the worst performing participants were removed thereby reducing generalizability.

The number of younger and older participants that were excluded is very similar (10 younger, 11 older) and due to a variety of reasons (e.g., failure to complete the study, fewer than 10 usable trials, medical/psychiatric history, etc.). However, we appreciate the reviewer’s concern about the disproportionate exclusion rate due to the number of correct trials. Our trial selection criteria was conservative, which likely contributed to the higher exclusion rate. The task was designed to keep participants at their estimated memory span, and participants needed to achieve 10 correct trials that remained after preprocessing.

We have added the following caveat to the limitations section of the revised manuscript:

“Another consideration is that our sample was generally high-performing, and participants who did not achieve 10 correct trials were removed from analysis (n=12, 8 older), which may limit generalizability to older adults with poorer cognitive performance.”

Relatedly, were any efforts taken to ensure that the older and younger adult samples were matched on variables other than age? For example, given that the younger adults were college students, were the older adults college graduates?

We did not take any measures to match participants during recruitment but we do report in the manuscript that only one of the older adults scored below the cutoff of 25 on the MocA, indicating the general absence of cognitive impairment in our sample of older adults. We agree with the reviewer that information about educational attainment in the older adult sample could be beneficial to readers. We have added that information to the Participants subsection of the Method in the revised manuscript (see added sentence below).

“All but one of the older adults in the sample had attained at least an Associates degree at the time of their participation in this study and 35% (N = 7; Masters = 6, Doctorate =1) had attained a graduate degree.”

I am concerned about the use of visual inspection for rejecting bad epochs: "The data were then visually inspected for extreme artifacts. Individual channels containing excessive artifact were noted and epochs containing excessive artifact across channels were removed from the continuous data." My worry is that this introduces too much subjectivity. Most studies use some objective definition to define an epoch as bad such as voltage deviations above some number of standard deviations. At a minimum I'd like more information on who inspected the data: were they blind to the purposes of the experiment? Were they blind to the age group of the subject? Were they blind to the condition of the epochs? But really, I think a more rigorous and objective method of detecting bad data should be used.

We appreciate the reviewer’s concern and hope that the clarification below will be helpful. As is described in the methods, the visual rejection of bad data was used only during the ocular artifact removal (training the ICA). The sentences on page 16, line 347-348 of the manuscript describe that “The retained independent components were then back-projected to the raw, downsampled (1000 Hz) data (i.e., 65-channel electrode space).” In other words, once the ICA is trained using the filtered (1-75Hz) data with large artifacts removed (by visual inspection) the resulting ICA decomposition is applied to a copy of the recordings without any data having been rejected. This procedure follows all of the recommendations of Viola et al. (2010) for optimal performance of ICA for ocular artifact removal.

As the reviewer suggests, an objective voltage threshold is typically used to identify artifactual segments during preprocessing and that is the case in this study. On page 17, lines 356-357 of the submitted manuscript, it is stated that “Bad channels within each data segment were identified using a maximum-minimum voltage threshold of 200 μV.”

Viola, F. C., Debener, S., Thorne, J., & Schneider, T. R. (2010). Using ICA for the analysis of multi-channel EEG data. Simultaneous EEG and fMRI: Recording, Analysis, and Application: Recording, Analysis, and Application, 121-133.

Line 358: how many segments were excluded due to >10% bad channels?

We have edited the sentence on Line 358 to include the average number of segments removed (see below).

“Bad channels within each data segment were identified using a maximum-minimum voltage threshold of 200 μV. Finally, segments with more than 7 (10%) channels marked as bad were excluded from the analysis, resulting in the removal of an average of one trial (SD = 2.96) per condition.”

What are the error bars on Figure 2?

We thank the reviewer for pointing out that we neglected to mention this in our original submission. The error bars in Figure 2 represent the standard error of the mean. We have added a statement to the caption for Figure 2 indicating that this is the case.

Finally, the data for the manuscript are not easily available. The authors state “Data are available from the corresponding author upon request.” but this seems to violate the PLOS statement that “Stating ‘data available on request from the author’ is not sufficient. If your data are only available upon request, select ‘No’ for the first question and explain your exceptional situation in the text box.”

We apologize for the oversight and will indicate upon resubmission that all data will be made publicly available.

---

## [Decision Letter · Decision Letter 1]

21 Dec 2023

PONE-D-23-16543R1Event-related Theta and Gamma Band Oscillatory Dynamics During Visuo-spatial Sequence Memory in Younger and Older AdultsPLOS ONE

Dear Dr. Kieffaber,

Thank you for submitting your manuscript to PLOS ONE. After careful consideration, we feel that it has merit but does not fully meet PLOS ONE’s publication criteria as it currently stands. Therefore, we invite you to submit a revised version of the manuscript that addresses the points raised during the review process.

We look forward to receiving your revised manuscript.

Kind regards,

Vilfredo De Pascalis

Academic Editor

PLOS ONE

Journal Requirements:

Additional Editor Comments:

I invite the authors to address the methodological questions raised by Reviewer 2 before the second manuscript revision is reconsidered for publication.

The authors are invited to provide a point-by-point response to each raised question.

Reviewers' comments:

Reviewer's Responses to Questions

**Comments to the Author**

1. If the authors have adequately addressed your comments raised in a previous round of review and you feel that this manuscript is now acceptable for publication, you may indicate that here to bypass the “Comments to the Author” section, enter your conflict of interest statement in the “Confidential to Editor” section, and submit your "Accept" recommendation.

Reviewer #1: All comments have been addressed

Reviewer #2: (No Response)

2. Is the manuscript technically sound, and do the data support the conclusions?

Reviewer #1: Yes

Reviewer #2: Yes

3. Has the statistical analysis been performed appropriately and rigorously? 

Reviewer #1: Yes

Reviewer #2: Yes

4. Have the authors made all data underlying the findings in their manuscript fully available?

Reviewer #1: Yes

Reviewer #2: Yes

5. Is the manuscript presented in an intelligible fashion and written in standard English?

Reviewer #1: Yes

Reviewer #2: Yes

6. Review Comments to the Author

Reviewer #1: No further comments

No further comments

No further comments

No further comments

No further comments

No further comments

Reviewer #2: The revisions have addressed all of my original concerns. One pice of new information provided, however, raises a new concern. Specifically in responses to my request "how many segments were excluded due to >10% bad channels" the authors respond "...Finally, segments with more than 7 (10%) channels marked as bad were excluded from the analysis, resulting in the removal of an average

of one trial (SD = 2.96) per condition." My concern is that if the average was 1 but the SD was almost 3, then the distribution must be quite skewed across conditions? Is this accurate? And if so, does differential exclusion rates across conditions introduce a confound? All I'm asking for here is a little more detail (e.g., give the number per condition) and some comment on if/how it affects the interpretation of the data.

7. PLOS authors have the option to publish the peer review history of their article (what does this mean?). If published, this will include your full peer review and any attached files.

Reviewer #1: **Yes: **CARLOS M. GOMEZ

Reviewer #2: No

---

## [Author Response · Author response to Decision Letter 1]

3 Jan 2024

Reviewer #1: 

No Comments

Reviewer #2: 

One pice of new information provided, however, raises a new concern. Specifically in responses to my request "how many segments were excluded due to >10% bad channels" the authors respond "...Finally, segments with more than 7 (10%) channels marked as bad were excluded from the analysis, resulting in the removal of an average of one trial (SD = 2.96) per condition." My concern is that if the average was 1 but the SD was almost 3, then the distribution must be quite skewed across conditions? Is this accurate? And if so, does differential exclusion rates across conditions introduce a confound? All I'm asking for here is a little more detail (e.g., give the number per condition) and some comment on if/how it affects the interpretation of the data.

First, we appreciate the reviewer’s concern with respect to the potential that varying the number of trials across conditions would introduce a potential confound. Fortunately, this concern is mitigated by the systematic selection of equal numbers of trials across all conditions for the analysis. This systematic selection is described at the bottom of page 17 in the original manuscript.

Second, we realized that reporting the variance across conditions is somewhat unusual and that reporting the descriptive statistics across subjects would be the more conventional way to report on the rejection of data. As a result, we have modified the description to include descriptive information over subjects and an analysis of variance to describe variability across conditions. Below is the revised description of data rejection that can be found in the middle of page 17 of the revised manuscript.

“Finally, segments with more than 7 (10%) channels marked as bad were excluded from the analysis, resulting in the removal of an average of 1.0 (SD = 2.2) trials per subject across all conditions. The number of trials rejected did vary significantly across the three encoding conditions, F(2,116) = 5.44, p < 0.01, but did not significantly differ across the three recall conditions, F(2, 116) = 2.23, p = 0.11. The potential confound introduced by varying the number of trials across sequence positions was mitigated by the systematic selection of an equal number of trials (described below).”

---

## [Editor Report · Decision Letter 2]

17 Jan 2024

Event-related Theta and Gamma Band Oscillatory Dynamics During Visuo-spatial Sequence Memory in Younger and Older Adults

PONE-D-23-16543R2

Dear Dr. Kieffaber,

We’re pleased to inform you that your manuscript has been judged scientifically suitable for publication and will be formally accepted for publication once it meets all outstanding technical requirements.

Kind regards,

Vilfredo De Pascalis

Academic Editor

PLOS ONE

Additional Editor Comments (optional):

I see that the authors have addressed the minor changes. Thus, it can be accepted for publication.

I apologize for the long time it took to conclude the review process. I want to congratulate the authors on the excellent quality of their research work.
---

## [Editor Report · Acceptance letter]

22 Mar 2024

PONE-D-23-16543R2 

PLOS ONE

Dear Dr. Kieffaber, 

I'm pleased to inform you that your manuscript has been deemed suitable for publication in PLOS ONE. Congratulations! Your manuscript is now being handed over to our production team.

Kind regards, 

on behalf of

Prof. Vilfredo De Pascalis 

Academic Editor

PLOS ONE